# Revisiting Protein Quality Assessment to Include Alternative Proteins

**DOI:** 10.3390/foods11223740

**Published:** 2022-11-21

**Authors:** Efrat Monsonego Ornan, Ram Reifen

**Affiliations:** Institute of Biochemistry and Nutrition, The Robert H. Smith Faculty of Agriculture, Food and Environment, The Hebrew University of Jerusalem, Rehovot 7610001, Israel

**Keywords:** longitudinal growth, bone development, diet, amino acids

## Abstract

The high demand for novel and existing sustainable protein sources (e.g., legumes, insects, algae, and cultured meat) to replace the animal-based sources is becoming crucial. This change in protein consumption calls for the re-evaluation of the current methods to assess their quality and bioavailability. The two conventional scores for PDCAAS (protein digestibility-corrected AA score) and DIAAS (Digestible Indispensable AA Score) have their limitations and have not been re-evaluated and updated to address plant and novel proteins’ quality. We suggest a sensitive physiological preclinical model that can rapidly and confidently address proteins from different sources. Our model is based on the postnatal growth, a major parameter for development and health in children, that influenced by environmental nutritional and lifestyle factors. Our results demonstrate that, with an appropriate amount of protein in the diet, almost all tested proteins performed as well as casein, the animal source. However, upon restriction (10% of calories), all alternative sources did not accomplish normal growth performance. Surprisingly, when compared to PDCAAS and DIAAS parameters obtained from the literature, no correlations were found between growth performance and these parameters, demonstrating their limitations.

## 1. Introduction

According to the projections by the United Nations (UN), the global population will increase by more than one billion people within the next 15 years, reaching 8.5 billion in 2030 and 9.7 billion in 2050 [1]. These trends leads to increase demand for food and especially in the fundamental and crucial macro-nutrient, proteins [2,3]. 

Protein is an indispensable energy source for the human body, crucial for tissue growth and essential living functions. Today, 40% of the global protein intake is from animal sources, while in Europe, this number rises to 67% (FAO (2020)). Yet, excessive animal protein consumption ballasts human health (e.g., leading to obesity, diabetes, and cardiovascular diseases) and has negative environmental impacts. Animal proteins require approximately 100-fold more water than that required to produce the same amount of plant protein [4]; the expanded production of livestock, livestock feed, and livestock products put biodiversity at risk; introduce air water and soil pollutants, as well as pesticides and antibiotics in the food chain; and animal production also increases the emission of greenhouse gasses, degrades grazing areas, and increases exposure to infectious diseases [5,6]. 

In light of the growing demand for protein, both novel and existing sustainable protein sources (e.g., legumes, insects, algae, and cultured meat) are becoming indispensable [4,7,8]. Consumer demand for alternative protein-based products is high and expected to grow considerably in the next decade. Factors contributing to the rise in the popularity of alternative plant proteins include: (1) potential health benefits associated with increased intake of plant-based diets; (2) consumer concerns regarding adverse health effects of consuming diets high in animal protein (e.g., increased saturated fat); (3) consumer recognition of the need to improve the environmental sustainability of food production; (4) ethical issues regarding the treatment of animals; and (5) general consumer view of protein as a “positive” nutrient (more is better).

## 2. Determination of Protein Quality

This dramatic change in protein consumption in general and specifically in plant proteins calls for re-evaluation of the current methods to assess their quality and bioavailability. Is there one solution that fits all? Are the methods of evaluating animal proteins relevant to assess plant proteins? The requirement thus far for a protein to be considered high quality, or complete, for humans is adequate levels of essential amino acids to support human growth and development. Various methods for evaluating protein quality have been developed over the years.

The protein digestibility-corrected amino acid score (PDCAAS) has been adopted by the FAO/WHO as the preferred measurement for protein value in human nutrition. The method is based on comparison of the first limiting essential AA concentration in the tested protein with the concentration of that AA in a reference pattern. This scoring pattern is derived from the essential AA requirements of preschool-age children. The chemical score obtained is corrected for the true fecal digestibility of the tested protein. PDCAAS values higher than 100% are truncated to 100%. The method has been questioned mostly because of a lack of a strong validity of correction for fecal instead of ileal digestibility and the truncation of PDCAAS values [9,10].

The scoring pattern does not include conditionally obligatory AA, which also contribute to the nutrition value of a protein. There is strong evidence that ileal, and not fecal, digestibility is the right parameter for correction of the AA score. The use of fecal digestibility overestimates the nutritional value of a protein, because AA nitrogen entering the colon is lost for protein synthesis in the body and is excreted in urine as ammonia. The truncation of PDCAAS values to 100% can be justified only for a limited number of situations in which the protein is the sole source of AA in the diet. An often-neglected aspect of plant proteins is their high content of important dispensable/conditionally indispensable AA [11]. Soy protein, for example, is higher in arginine [8].

The Digestible Indispensable AA Score (DIAAS) is defined as: DIAAS% = 100 × [(mg of digestible dietary indispensable AA in 1 g of the dietary protein)/(mg of the same dietary indispensable AA in 1 g of the reference protein)]. In March 2013, the FAO has endorsed the DIAAS to replace the previously recommended PDCAAS for protein quality assessment [12]. The DIAAS accounts for AA digestibility at the end of the small intestine, providing a more accurate measure of the amounts of AA absorbed by the body and the protein’s contribution to human AA and nitrogen requirements, in contrast to PDCAAS, which is based on an estimate of digestibility over the total digestive tract.

DIAAS formulation centered on fast-growing animal models rather than humans, and a focus on individual isolated foods vs. the food matrix. Yet, there are substantial limitations, many of which are accentuated in the context of a plant-based dietary pattern, including a failure to translate differences in nitrogen-to-protein conversion factors between plant- and animal-based foods, the limited representation of commonly consumed plant-based foods within the scoring framework, and the inadequate recognition of the increased digestibility of commonly consumed heat-treated and processed plant-based foods. The DIAAS is also increasingly used out of context where its application could produce erroneous results, such as exercise settings. When investigating protein quality, particularly in a plant-based dietary context, the DIAAS should ideally be avoided [13,14]. In this report we present a different approach to evaluate protein quality and compare it to the available scores. We suggest that this new method to assess the health implications of the diet is more suitable for the novel and future alternative proteins.

## 3. Results: A New Way to Assess Protein Quality

The use of novel protein sources requires a more sensitive and accurate comparison between the novel and existing proteins, in terms of their health impact during the life cycle. We suggest a sensitive physiological preclinical model that can rapidly and confidently address the health issue of proteins from different sources. Our model is based on the postnatal growth, defined as the gain in body size and weight during the juvenile period. Postnatal growth is the major parameter for development and health in children and is heavily influenced by environmental factors, such as nutrition and lifestyle. Skeleton development is heavily influenced by the consumption of various compounds during the period of rapid growth. Thus, to maximize growth and peak bone mass, balanced nutrition and general health must be optimal before the onset of puberty and maintained throughout this period until puberty [15,16,17].

The preclinical model of skeleton development in young rats is especially suitable to address minute modifications in the diet’s components for a few reasons. (a) The rats at this age are extremely sensitive to any deficiency due to their accelerated growth rate, and the skeleton is particularly reactive to these changes and also represents further developmental sequences in the body. (b) The whole period of juvenile growth, from post-weaning to sexual maturation lasts for 10 weeks (in comparison to 10–15 years in human), allowing timely preclinical test to conclude the effects of the diet and plan strategy before progressing to clinical studies. (c) The rat model is highly acceptable as a preclinical model to study nutrition and growth, due to similarities in nutritional needs and developmental processes.

The caloric intake of the rats in the protein deficient groups were lower than the PD-Cas group that did not differ from the Ctrl group which also consumed casein but in the recommended amounts. In line with this, the growth performance (total weight and total longitudinal growth) of the two groups that consumed casein as their protein source did not differ, suggesting that the half amount of casein was sufficient to support normal growth. Even though rats from the other protein-deficient diet groups consumed less energy and food then the casein groups: PD-Fly and PD-CP/F consumed moderate amounts of food and energy, while PD-CP/I, PD-Spl, and PD-Soy consumed the lowest. The growth performance of these groups correlates to their food consumption; as the rats from the Ctrl and PD-Cas groups had the significantly highest weight and length, the PD-Fly and PD-CP/F groups and then the rats from PD-CP/I, PD-Spl, and PD-Soy groups showed significantly lighter weights and the shorter lengths (Figure 1A—10%). Interestingly, when the amount of protein was as recommended (Figure 1B—20%), all groups consumed higher amounts of protein as compared to casein (Ctrl), and except for the spirulina-consuming group, performed as well as the Ctrl group in terms of growth, i.e., body weight and length.

## 4. Discussion and Summary

Our results demonstrate that, with an appropriate amount of protein in the diet, almost all tested proteins performed as well as casein, that serves as the reference for longitudinal growth. The only exception was spirulina. However, upon restriction (10% of calories), all alternative sources did not accomplish normal growth performance. Surprisingly, when compared to PDCAAS and DIAAS parameters obtained from the literature, no correlations were found between theses parameters and growth performance, demonstrating the limitations of the common scores for protein quality. For instance, soy as a protein source (10%), which was expected to be almost comparable with casein according to PDCASS and DIAAS, achieved very low growth performance, while chickpea flour, with its lower parameters, performed better in terms of weight and length.

In light of the increased use of plant proteins and the limitations of the existing methods to evaluate protein quality and digestibility in general and specifically in plant protein, we suggest a new simple and a more physiologically accurate way to assess protein quality. This method also allows the analyses of not only the effect of pure proteins but also the effects of processing methods, amounts in the diet, and combinations of various sources. All in all, the model provides a tool with which to study the real physiological impact of novel dietary protein.

### Method: The Experimental Model:

The animals: Female Sprague Dawley rats after weaning (3 weeks old) were randomly separated into groups (*n* = 8 for each). All groups were treated with a semi-purified diets based on the American Institute of Nutrition (AIN-93G) recommendations for rodents in the growth phase. All rats had ad libitum access to food and liquids. To evaluate the influence of the different sources of protein on growth, a 6-week long experiment was conducted on rats after weaning. This time frame was selected in order to mimic the human growth period up to sexual maturity.

The diets: The diet modifications were the source of protein and the relative energy contribution of each macronutrient. The rats from all groups consumed casein (Ctrl/Cas) as a standard protein source, while one of the diets contained the recommended amount for protein and the other was deficient in protein. The other experimental groups consumed standard or protein-deficient (PD) diet, whereas the protein source was either from soy isolate (soy), spirulina powder (Spl), chickpea isolate (CP/I), chickpea flour (CP/F), or fly larvae protein powder (Fly). The tested diets were either deficient in protein (10% protein, 60% carbohydrate, 30% of the caloric intake) or standard (20% protein, 60% carbohydrate, 20% fat of the caloric intake) and supplemented with sufficient amounts of vitamins and minerals.

The tested parameters: Throughout the experiment, food body weight (g), and length (cm) were monitored on a weekly basis. Body weight represents the growth rate of the whole body, including energy balance. Body length, from tip to nose, is considered an indication for longitudinal bone growth [18,19].

## Figures and Tables

**Figure 1 foods-11-03740-f001:**
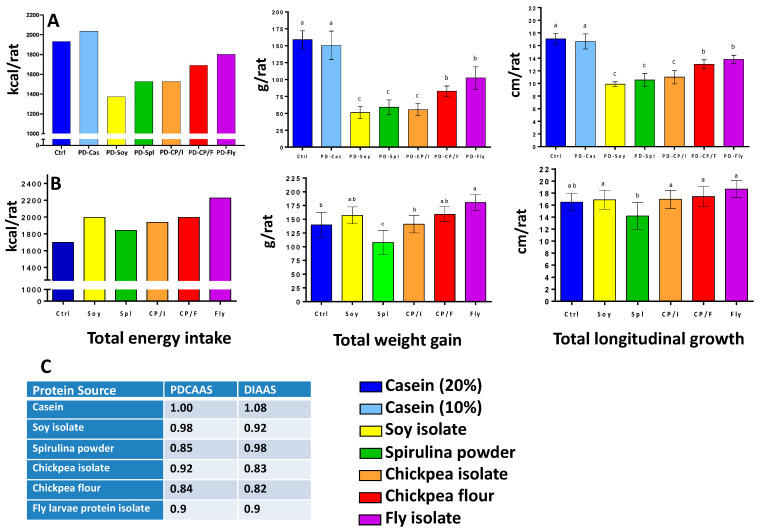
Energy Consumption and Growth Patterns. Total energy intake, weight gain, and longitudinal growth of rats in the 6-week experimental period. Comparison of diets with 10% protein (**A**); Comparison of diets with 20% protein (**B**); Published PDCASS and DIASS parameters for these proteins [10,11,12,13,14] (**C**). Abbreviations: Ctrl—20% casein; PD-Cas—10% casein; Soy—20% soy isolate; PD-Soy—10% soy isolate; Spl—20% spirulina powder; PD-Spl—10% spirulina powder; CP/I—20% chickpea isolate; PD-CP/I—10% chickpea isolate; CP/F—20% chickpea flour; PD-CP/F—10% chickpea flour; Fly—20% fly larvae protein isolate; PD-Fly—10% fly larvae protein isolate. Values are expressed as mean ± SD of *n* = eight rats/group, different superscript letters are significantly different (*p* < 0.05) by one-way ANOVA followed by Tukey’s test.

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
