# Peer review of "Revisiting Protein Quality Assessment to Include Alternative Proteins"

_foods, 2022, doi:10.3390/foods11223740_

Round 1
Reviewer 1 Report
A sensitive physiological preclinical model author proposed is useful for evaluate protein in the diet. My major comments:
1.The detailed model paprameters author established need to be provided. Readers can not understand the model establishment method in present form.
2.What samples or patients could be carried out through this model, namely, the application of this model.
Author Response
Reviewer 1
A sensitive physiological preclinical model author proposed is useful for evaluate protein in the diet. My major comments: Thanks
- The detailed model paprameters author established need to be provided. Readers can not understand the model establishment method in present form.
The methodological details of the model are described in details in the section named Method: The experimental model. The paragraph is situated after the result section as requested by the journal. In order to make this part more comprehensive to follow the reviewer’s comments it was edited and separated into subtitles. The animals; the diets; the tested parameters. This model is based on previous studies from our lab in which the rat model served as pre-clinical model for the effect of different diets on growth parameters.
Sides, R.; Griess-Fishheimer, S.; Zaretsky, J.; Shitrit, A.; Kalev-Altman, R.; Rozner, R.; Beresh, O.; Dumont, M.; Penn, S.; Shahar, R.; Monsonego-Ornan, E. The Use of Mushrooms and Spirulina Algae as Supplements to Prevent Growth Inhibition in a Pre-Clinical Model for an Unbalanced Diet. Nutrients 2021, 13, 4316. https://doi.org/10.3390/nu13124316
Griess-Fishheimer, S.; Zaretsky, J.; Travinsky-Shmul, T.; Zaretsky, I.; Penn, S.; Shahar, R.; Monsonego-Ornan, E. Nutritional Approaches as a Treatment for Impaired Bone Growth and Quality Following the Consumption of Ultra-Processed Food. Int. J. Mol. Sci. 2022, 23, 841. https://doi.org/10.3390/ijms23020841
Zaretsky, J., Griess-Fishheimer, S., Carmi, A. et al. Ultra-processed food targets bone quality via endochondral ossification. Bone Res 2021 9, 14. https://doi.org/10.1038/s41413-020-00127-9
Travinsky-Shmul, T.; Beresh, O.; Zaretsky, J.; Griess-Fishheimer, S.; Rozner, R.; Kalev-Altman, R.; Penn, S.; Shahar, R.; Monsonego-Ornan, E. Ultra-Processed Food Impairs Bone Quality, Increases Marrow Adiposity and Alters Gut Microbiome in Mice. Foods 2021, 10, 3107. https://doi.org/10.3390/foods10123107
In these papers other diets were used (not related to protein source), and several parameters were further investigated. Here, in this report we harness the model to the protein question.
- What samples or patients could be carried out through this model, namely, the application of this model.
The relevancy of the model is of great importance. We decided to use the Postnatal growth in the rat model since it is the major parameter for growth development and health in children, and heavily influenced by environmental factors such as nutrition and life style. Indeed, to maximize growth and peak bone mass, balanced nutrition and general health must be optimal before the onset of puberty and maintained throughout this period until puberty. Thus, the model is relevant to the young population in the period of the post embryonic developmental stage, from age of 2 years to adolescence. This issue was further emphasized in the text.

Reviewer 2 Report
As the protein digestibility-corrected amino acid score- PDCAAS and the Digestible Indispensable AA Score- DIAAS have limitations, the manuscript established a sensitive physiological preclinical model that can rapidly and confidently address proteins from different sources. I have some comments, as shown below.
1. Please be precise for preclinical model. Preclinical model means human trial. However, the author used animal model not the human trial.
2. Does animal model can reflect the real digestion and absorption of proteins for human?
3. I don’t think the weight gain is positively correlated with the protein quality. Weight gain is linked with the amount of nutrients, including carbohydrate, fat, protein and vitamins. I would like to know how is the nutrients composition of fly group? What is the normal diet of the rat?The high amount of weight gain in figure 1A and 1B may be related to other composition consumption.
4. Please provide figure number and figure legends in the manuscript. Why the groups were not the same for 10% protein and 20% protein?
5. Based on the new evaluation method which indicator is the most important for protein quality?total weight gain or longitudinal growth? What is the standard for the evaluation?
6. What is the correlation result between the new method and PDCAAS, DIAAS? Could the author provide the results?
7. In general,I think the results are very few and not much discussion on the advantage of the new methods.
8. The font in the abstract should be consistent
9. Line58: based on comparison of the of the first limiting essential AA, please correct of the of
10. The language still needs to be improved
Author Response
Comments and Suggestions for Authors
As the protein digestibility-corrected amino acid score- PDCAAS and the Digestible Indispensable AA Score- DIAAS have limitations, the manuscript established a sensitive physiological preclinical model that can rapidly and confidently address proteins from different sources. I have some comments, as shown below.
Please be precise for preclinical model. Preclinical model means human trial. However, the author used animal model not the human trial. By preclinical model we meant animal studies that mimic the clinical set up and will enable us to decide how to further study these queries in the clinical set (human study). This is the NIH definition: Research using animals to find out if a drug, procedure, or treatment is likely to be useful. Preclinical studies take place before any testing in humans is done.
- Does animal model can reflect the real digestion and absorption of proteins for human?
This is very important issue and we discuss it in this report as well as cite other papers that deal with the issue. Rodents model is very acceptable in the study of mutations, disease and drugs as well as nutrition. Thus we suggest that it can reflect and mimic human developmental questions. It is important to say that these type of studies are close to impossible to execute in humans and therefore this is probably the closest model to humans that can be used
- I don’t think the weight gain is positively correlated with the protein quality. Weight gain is linked with the amount of nutrients, including carbohydrate, fat, protein and vitamins. I would like to know how is the nutrients composition of fly group? What is the normal diet of the rat?The high amount of weight gain in figure 1A and 1B may be related to other composition consumption.
Certainly, we agree with this comment, protein is only one parameter, and our results show that in different composition of diet the weight results differ. To overcome these issues we planed the diets to be isocaloric and equal in the amount of macro and micro- nutrients except for the protein source, thus we can compare between rats and connect it to the specific protein.
In the explanation about the diets in the section Method: The experimental model it is written: The diets modifications were the source of protein and the relative energy contribution of each macronutrient. The rats from all groups consumed casein (Ctrl/Cas) as standard protein source…. The other experimental groups consumed diet, whereas the protein source was either from soy isolate (Soy), spirulina powder (Spl), chickpea isolate (CP/I), chickpea flour (CP/F) or fly larvae protein powder (Fly). The tested diets were either deficient in protein (10% protein, 60% carbohydrate, 30% at of the caloric intake), or standard (20% protein, 60% carbohydrate, 20% fat of the caloric intake) and supplemented with sufficient amounts of vitamins and minerals.
- Please provide figure number and figure legends in the manuscript. Why the groups were not the same for 10% protein and 20% protein?
Since this is a brief report we use only one figure the figure number was added. And the explanation for the parts of the figure are now in the legend. Stating that 1A is diet with 10% protein in the diet while 1B is 20% protein in the diet. These differences in the diet and in the checked parameters show the importance of preclinical model in contrast to the PDCAAS and DIAAS parameters.
- Based on the new evaluation method which indicator is the most important for protein quality? total weight gain or longitudinal growth? What is the standard for the evaluation?We truly can not point on a single parameter; this is one of the conclusions in our report. We believe that protein quality is way more complicated to be summarized in one number, this is our criticism on the present scores. As we wrote ”This change in protein consumption calls for reevaluation of the current methods to assess their quality and bioavailability. The protein digestibility-corrected amino acid score- PDCAAS and the Digestible Indispensable AA Score- DIAAS, have their limitations, and were not re-evaluated and updated to address plant and novel proteins’ quality.
- 6. What is the correlation result between the new method and PDCAAS, DIAAS? Could the author provide the results? We show these results in the figure (not a good correlation) and mention in the text “Surprisingly, when compared to PDCAAS and DIAAS parameters obtained from the literature, no correlations were found between growth performance and these parameters, demonstrating their limitations. For instance, Soy as protein source (10%), that was expected to be almost comparable with casein according to PDCASS and DIAAS, achieved very low performance of growth results”
- In general think the results are very few and not much discussion on the advantage of the new methods. As I mentioned, this is a brief report that cannot include a lot of details, but aim to open a discussion in our field regarding the way to conclude protein quality, and to discuss the limitations of the current methods. I am sure that more research in this line (from our group and other) will expand the understanding of this important issue.
- The font in the abstract should be consistent thank you this is corrected now
- Line58: based on comparison of the of the first limiting essential AA,please correct of the of Corrected
- The language still needs to be improved We went through all the manuscript to correct typos and language mistakes.

Reviewer 3 Report
Hi dear
This article "Revisiting protein quality scoring to include alternative proteins” was revised and has a novelty and I recommend consideration of the following comments.
Title: If you can rewrite and make it more interesting for readers. I propose: “Revising protein quality assessment to include alternative proteins”.
Abstract:
· “The protein digestibility-corrected amino acid score- PDCAAS and the Digestible Indispensable AA Score- DIAAS, have their limitations, and were not re-evaluated and updated” to address plant and novel proteins’ quality. Please rewrite and revise another time because It is vague and difficult to understand.
· Synchronize the font in through the abstract.
· AA Score- DIAAS & score- PDCAAS for the first time of expression please consider self-explanatory.
Keywords: Please choose keywords for your manuscript.
Abbreviation:
· Please provide “Abbreviation section consequent the Keywords
Introduction:
· Indispensable word is very repeatable. Please exchange it to synonym words.
· Line 42-44: (1) potential health benefits associated with increased intake of plant-based diets and (2) consumer concerns regarding adverse health effects of consuming diets high in animal protein (e.g., increased saturated fat). What is the difference between them?
· Please in the end of introduction express the aim of your study.
“Results:
· “3. Results: A new way to assess protein quality is needed” Please express as phrase not to ordinary sentence.
· Figure and Table do not title please consider it according to the scientific approach. In addition, the source used to express these figures and table should be mentioned. No statistical comparison has been made for “Total energy intake” within A-10% protein and B-20% protein why?
Discussion:
· Discussion text must grammar improve and in some cases it is very weak and maybe there is no discussion at all. Please strengthen:
· 3.5. Principal component analysis
Summary:
· 10% protein, 60% carbohydrate, 30% at of the caloric intake…. Please corrected it as the follow: 10% protein, 60% carbohydrate, 30% fat of the caloric intake….
· The summary ought to be comprehensive and concise in detail.
· Uniformity in font and line spacing should be observed.
References: It is OK.
The article has many flaws in express and concept of English, it is suggested to be revised in a scientific and native way.

Author Response
Comments and Suggestions for Authors
This article "Revisiting protein quality scoring to include alternative proteins” was revised and has a novelty and I recommend consideration of the following comments.
Title: If you can rewrite and make it more interesting for readers. I propose: “Revising protein quality assessment to include alternative proteins”. Thanks, we accept this suggestion.
Abstract:
- “The protein digestibility-corrected amino acid score- PDCAAS and the Digestible Indispensable AA Score- DIAAS, have their limitations, and were not re-evaluated and updated” to address plant and novel proteins’ quality.Please rewrite and revise another time because It is vague and difficult to understand. Corrected to: The two conventional scores for PDCAAS (protein digestibility-corrected amino acid score) and DIAAS (Digestible Indispensable AA Score), have their limitations, and were not re-evaluated and updated to address plant and novel proteins’ quality.
Synchronize the font in through the abstract. Corrected
- AA Score- DIAAS & score- PDCAAS for the first time of expression please consider self-explanatory.
Keywords: Please choose keywords for your manuscript.
Abbreviation: Please provide “Abbreviation section consequent the Keywords
These were corrected by adding keywords and abbreviations which includes AA, PDCAAS, DIAAS.
Introduction:
- Indispensable word is very repeatable. Please exchange it to synonym words. Corrected
- Line 42-44: (1) potential health benefits associated with increased intake of plant-based diets and (2) consumer concerns regarding adverse health effects of consuming diets high in animal protein (e.g., increased saturated fat). What is the difference between them? The difference is the objective vs. the subjective, the real nutritional values of the different proteins, and the believe of the consumer regarding health risk/benefit of protein.
Please in the end of introduction express the aim of your study. We added: In this report we present a different approach to evaluate protein quality, and compare it to the available scores. We suggest that this new method to assess the health implication of the diet is more suitable for the novel and future alternative proteins.
“Results:
- “3. Results: A new way to assess protein quality is needed” Please express as phrase not to ordinary sentence. Done
- Figure and Table do not title please consider it according to the scientific approach.In addition, the source used to express these figures and table should be mentioned. No statistical comparison has been made for “Total energy intake” within A-10% protein and B-20% protein why? The figure was corrected. The figures A and B are results from the animal studies done in our lab as described in the results and in M&M. the table of the PDCAAS and AIAAS is based on the literature, the relevant references were added to the legend. We do not present the statistics for food consumption since the results did not did pass the normality and homoscedasticity Bartlett’s test, thus we do not imply for statistically significant differences, but still present the averaged amount of calories consumed in the different groups as measured for the same experiment that reveal differences in body and length weight.
Discussion:
- Discussion text must grammar improve and in some cases it is very weak and maybe there is no discussion at all. Please strengthen: Done, although this is a brief report which is limited in words.
Summary:
- 10% protein, 60% carbohydrate, 30% at of the caloric intake…. Please corrected it as the follow: 10% protein, 60% carbohydrate, 30% fat of the caloric intake…. Corrected
- The summary ought to be comprehensive and concise in detail. Corrected
- Uniformity in font and line spacing should be observed. Corrected
References: It is OK.

Reviewer 4 Report
Unfortunately, the manuscript has not been well designed. It is poorly written. The content of the manuscript is not sufficient for publication in foods.
Author Response
This is a brief report, and should be considered accordingly
Round 2
Reviewer 2 Report
Although the author answered most of my question, however, there was still two questions not addressed yet.
1. As the pre-clinical model evaluation has very low correlation with PDCAAS and DIAAS, why the author think the new evaluation method is better than PDCAAS and DIAAS? Please discuss the details and provide evidence on this point.
2. I am still doubting only the weight gain and growth rate could not reflect the protein quality and bioavailability as other nutrients may also influence weight gain and growth rate.
Author Response
Thanks for your comments:
As the pre-clinical model evaluation has very low correlation with PDCAAS and DIAAS, why the author think the new evaluation method is better than PDCAAS and DIAAS? Please discuss the details and provide evidence on this point.
We suggest that the current methods are not satisfying, and show that straight forward method, that check the basic physiology of growing is better or at least could not be ignored. Post embryonic development as measured by height and weight is in fact the regular way to check children, so the fact that the results of these basic measurements differ from the PDCAAS and DIAAS are disturbing and raise a concern regarding these methods, this discussed in the paper.
I am still doubting only the weight gain and growth rate could not reflect the protein quality and bioavailability as other nutrients may also influence weight gain and growth rate.
We agree with this concern, and the model address the issue by preparing the exact some diet that differ only in the protein source, thus we can postulate that the differences stems for the protein and not other nutrients.
Reviewer 4 Report
I understand that it's a brief report. Brief reports are similar to original research articles in that they follow the same rigor, format, and guidelines but are designed for small-scale research. As I mentioned before, the quality of the manuscript is not sufficient for scientific publication. The quality and writing style of the manuscript did not improve after revision! Unfortunately, I'm not agreeing to publish in the current format.
Author Response
Thanks for your comment, we edited the writing and the figure quality.